# Embedded Sensor Systems in Medical Devices: Requisites and Challenges Ahead

**DOI:** 10.3390/s22249917

**Published:** 2022-12-16

**Authors:** Nerea Arandia, Jose Ignacio Garate, Jon Mabe

**Affiliations:** 1TEKNIKER, Basque Research and Technology Alliance (BRTA), 20600 Eibar, Spain; 2Department of Electronics Technology, University of the Basque Country (UPV/EHU), 48080 Bilbao, Spain

**Keywords:** embedded systems, medical devices, new product development, medical device regulation

## Abstract

The evolution of technology enables the design of smarter medical devices. Embedded Sensor Systems play an important role, both in monitoring and diagnostic devices for healthcare. The design and development of Embedded Sensor Systems for medical devices are subjected to standards and regulations that will depend on the intended use of the device as well as the used technology. This article summarizes the challenges to be faced when designing Embedded Sensor Systems for the medical sector. With this aim, it presents the innovation context of the sector, the stages of new medical device development, the technological components that make up an Embedded Sensor System and the regulatory framework that applies to it. Finally, this article highlights the need to define new medical product design and development methodologies that help companies to successfully introduce new technologies in medical devices.

## 1. Introduction

Embedded Sensor Systems have become the key element of the advances in medical devices; the high versatility they offer enables the development of new diagnostic and advanced monitoring devices for patients in both home and hospital environments.

The medical device industry is regulated by different national notified or regulatory bodies. Two of the world’s main regulatory bodies are the European Commission Directorate and US Food & Drugs Administration (FDA). Figure 1 shows the most important regulatory authorities around the world.

In the European Union, medical devices are regulated by harmonised health regulations. Any manufacturer who wants to put a device on the European market must go to a notified body to have its device assessed [1]. If it is considered approved, a certificate of conformity with the CE mark is emitted, which allows it to be sold in all the countries of the European Union [2].

According to the European Medical Device Regulation (MDR), the following are considered medical devices:

“Any instrument, apparatus, appliance, software, implant, reagent, material or other article intended by the manufacturer to be used, alone or in combination, for human beings for one or more of the following specific medical purposes: Diagnosis, prevention, monitoring, prediction, prognosis, treatment or alleviation of disease, diagnosis, monitoring, treatment, alleviation of, or compensation for, an injury or disability, investigation, replacement or modification of the anatomy or of a physiological or pathological process or state, providing information by means of in vitro examination of specimens derived from the human body, including organ, blood and tissue donations and which does not achieve its principal intended action by pharmacological, immunological or metabolic means, in or on the human body, but which may be assisted in its function by such means [...]”[3]

In contrast, the U.S. Food and Drug Administration (FDA) considers a medical device:

“An instrument, apparatus, implement, machine, contrivance, implant, in vitro reagent, or other similar or related article, including a component part, or accessory which is: (A) recognized in the official National Formulary, or the United States Pharmacopoeia, or any supplement to them, (B) intended for use in the diagnosis of disease or other conditions, or in the cure, mitigation, treatment, or prevention of disease, in man or other animals, or (C) intended to affect the structure or any function of the body of man or other animals, and which does not achieve its primary intended purposes through chemical action within or on the body of man or other animals and which does not achieve its primary intended purposes through chemical action within or on the body of man or other animals and which is not dependent upon being metabolized for the achievement of its primary intended purposes”[4]

In both cases, we can see that to discern whether a product is a medical device or not, it is necessary to define its intended use [5]. For example, smartwatches, which have become so common recently, depending on their intended use, can be considered medical devices. A smartwatch that measures the pulse or performs an electrocardiogram is considered a medical device only if the readings are used to perform medical actions such as diagnosing a disease or establishing a treatment [6]. On the other hand, with the same technology, this smartwatch is not considered a medical device if it is only used for informative purposes. For example, when a user is interested in measuring the pulse rate after a workout. This is the category for devices such as the Apple Watch [7].

The design and development of these systems require considering very strict regulations from the conception phase of the idea. In [8], a study is presented detailing the challenges for the development of medical devices according to FDA regulations. Currently, regardless of country or region, the design and development of medical devices are regulated by several standards that guarantee the quality of the devices and minimise the exposition risk to healthcare professionals and patients [9,10].

The regulations to be applied, as well as their interpretation, depends on the intended use of the medical device and the technologies used. For example, there are purely software devices [11], ones that include software and electronics, or purely mechanical devices. For all these cases, there are common and specific regulations as well [12].

Innovation in the medical sector is often driven by start-ups, which often have great ideas but lack experience in the development of medical devices in accordance with medical regulations. Inexperience in the sector, together with difficulty in identifying these regulations and translating them into technical requirements, results in the development of new innovative medical devices that are not always successful. Article [13] highlights the difficulty for start-ups to cope with the new regulation that applies to the development of medical devices. In addition, the author of [14] states that there is a risk of slowing down innovation because the new MDR requires more costly and high-quality testing. It also requires more technical documentation to comply with the new regulation.

Therefore, the goal of this article is to review the major challenges and requirements in the development of Embedded Sensor Systems for medical devices. To this end, Section 2 presents the context of this article, the innovation in Health Technology. In Section 3, the possibilities offered by embedded systems in the healthcare sector are introduced. Then, in Section 4, the design and development process of a medical device is outlined. In Section 5, the technological blocks that compose an Embedded Medical Sensor System are described and in Section 6, the regulations that apply to each of the technological blocks are presented. Finally, Section 7 states the conclusions of this work and some insights about future lines of work.

There are several articles where technical and regulatory solutions are partially discussed, for example, in [15]. The author reviews the main regulatory challenges that a wearable sensor faces in order to be considered a medical device. Although the main regulations are discussed, the author does not go into detail on the main requirements of all of them. Likewise, in [16], different challenges associated with injectable hydrogels are presented. Emphasis is placed on the technical challenges, but no specific regulatory requirements are discussed. In [17], the author presents different technologies for measuring parameters related to sleep-disordered breathing. However, this article does not analyse the technical solutions from a regulatory point of view. Rather, it focuses on a technical comparison of measurement solutions. Similarly, in [18], the existing sensor technologies to be integrated into wearable solutions are reviewed.

This article, in contrast, aims to delve deeper into the technical and regulatory aspects that embedded sensor solutions must comply with to be compliant with medical product requirements. To this end, this article focuses on identifying those key points that designers of medical devices with Embedded Sensor Systems must consider. It also seeks to identify challenges and requirements that are common to embed medical devices and not specific to a particular solution or application.

## 2. Innovation in Health Technology

The medical device industry has constantly been evolving over the last few years. On the one hand, new healthcare challenges are emerging, such as COVID-19 or the problem of an elderly society [19]. On the other hand, the rapid evolution of technology is making it possible to improve current medical devices and solutions.

COVID-19, a disease caused by the new coronavirus known as SARS-CoV-2, emerged on 31 December 2019 in Wuhan (China) [20]. In spite of becoming a global pandemic, COVID-19 also introduced technological innovation into the healthcare sector. Tele-medicine has come to stay; before COVID-19, it was usual to go to the doctor face-to-face [21]. However, due to the collapse of health systems, mobile applications and information systems for patient care have been developed around the world [22]. Likewise, disinfecting robots [23], devices for monitoring temperature in public spaces [24] or low-cost oximeters for home use [25] are clear examples of the technological evolution that this pandemic has brought about.

The problem of an ageing population is another challenge to be faced. The increase in life expectancy and the considerable decrease in the birth rate make it essential to take measures to help manage and optimise patient care. The WHO (World Health Organisation) estimates that between 2015 and 2050, the world’s population over the age of 60 will increase from 12% to 22% [26]. In this context, technological developments oriented towards patient monitoring in both home and hospital environments are of special relevance.

Not only the emergence of new challenges in the sector has brought new technological innovations. The development of technology in aspects such as the Internet of Things (IoT) and Artificial Intelligence (AI) also makes it possible to develop a multitude of innovative Medical Technologies [27]. As a result, new solutions and devices have appeared in the healthcare sector that allow (i) the prevention of diseases or damages [28,29], (ii) the diagnosis of diseases or special conditions [30,31], (iii) the monitoring of the patient’s condition [32,33], (iv) helping treat and overcome diseases [34,35] and (v) caring for and facilitating the process of patient recovery [36,37].

Some of the advances and developments that are revolutionising the medical sector include:Optimisation of data management: an increasing number of mobile or portable devices are being used to monitor all kinds of patient parameters, temperature, oxygen, pulse rate, etc. [38]. Storing, organising and analysing all this data is not an easy task. However, thanks to Big Data and AI, it is possible to manage huge volumes of data in an efficient way [39,40]. The wearable system proposed in [41] is able to diagnose diabetes using machine learning and big data. In [42], a big data system is developed to support the rehabilitation of strokes and lung diseases. The heterogeneity of data capture systems leads to the development of architectures to support such different solutions. In [43], a semantic big data architecture to address the heterogeneity of data between different wearable platforms is presented.Artificial vision systems: these systems are a great enabler for the development of new systems in the field of healthcare. Complex systems, such as endoscopes, radiology, ophthalmology, surgery, etc., use this technology. Moreover, embedded vision, which is based on the integration of adapted camera modules that are directly incorporated into medical devices, enables intelligent image processing in a variety of portable applications. One of them are eye-tracking systems, which can be used for diagnostics or patient care. In [44], a vision algorithm is presented to detect eye movement for the identification of ocular pathologies, such as strabismus. In [45], an algorithm is presented that, used together with the Irisbond eye-tracker [46], is able to assess mathematics in children with cerebral palsy. Eye-tracking systems can even be useful for healthcare professionals. In [47], the possible use of eye-detection systems to assist neonatal resuscitation processes is presented. In [48], a pilot study is presented for the same purpose.Early diagnosis of diseases: in a few years, artificial intelligence will make it possible to diagnose diseases such as lung cancer [49]. The analysis of thousands of digital scans will identify early stages of cancer that would not have been possible with traditional technology [50]. In [51], different AI algorithms used for the diagnosis and treatment of prostate cancer are reviewed. Similar analyses are also supported by [52] for colorectal cancer detection and [53] for breast cancer detection.Patient monitoring at home and in the hospital: vital signs monitoring allows the patient’s progress to be evaluated and ensures early detection of undesirable effects. Advances in embedded electronics with integrated sensorisation allow reliable measurements of temperature, oxygen, pulse or blood pressure using comfortable, self-powered devices. More and more work is being performed on the development of wearable solutions, such as smartwatches, that allow continuous patient monitoring in a non-invasive way [54]. Article [55] presents a system capable of measuring heart rate, SpO2 and respiratory rate. It is a low-cost system, which makes it interesting for deployment in low-resource settings. There are also solutions in the literature that are capable of detecting falls [56,57]. These systems are especially interesting for elderly or very fragile patients [58].

In this context, the Research and Development (R&D) strategies of leading medical companies indicate the need to evolve and develop the current technology. For this purpose, many of them have alliances with universities and research centres in which they invest a large percentage of their annual revenues. Roche, the world’s largest biotech company, with revenues of about 65 billion Swiss francs in 2021 [59], invests around 9 billion Swiss francs in R&D every year [60], one of the highest innovation spending figures across all sectors. Medtronic, a leading manufacturer of medical technologies whose portfolio includes infusion pumps, medical devices and advanced electrical instrumentation for surgery, with a revenue of about USD 30 billion [61], invests more than USD 2.5 billion in R&D each year [62]. Other leading companies in the sector, such as Siemens [63] or Abbott [64], also invest around 10% of their turnover in R&D [65].

Medtronic launched the first patient procedure with the Hugo RAS robotic-assisted surgery system in 2021. This platform includes AI technology that records and processes images from the operating room [66]. Roche, through its collaboration with Microsoft since 2017, is transforming in vitro diagnostics with solutions based on the Microsoft Azure IoT Platform. In this way, Roche achieves intelligent and remote management of its in vitro devices [67]. Based on the success of this collaboration, in December 2021, Roche and Microsoft have signed a new agreement to integrate AI and cloud technology into their devices [68].

The evolution of technology, together with the multitude of programs promoted by the administrations to improve the healthcare of citizens, is generating a massive wave of start-up companies that aim to design, develop and put new medical devices on the market. Moreover, according to a report published by the Spanish Association of Business Angels Networks (AEBAN), 40% of the start-ups that have been created in Spain during the last few years belong to the medical sector [69]. It also explains that the most attractive sectors for Business Angels are mobility, health and energy.

Among the Spanish start-ups, Koa Health, Inbrain Neuroelectronics and MedLumics stand out as the start-ups with the most funding in 2021, with the three companies together totalling more than 60 million euros [70]. Koa Health works on a wide range of mental health solutions, from digital wellness to digital therapy, with the tools it offers aiming to improve the mental well-being of users anytime and anywhere [71]. Inbrain Neuroelectronics is developing a minimally invasive neural interface that can detect and modify specific biomarkers using AI and Big Data to help improve personalised neurological therapies [72]. MedLumics specialises in the development of cardiac optical imaging ablation devices for atrial treatment [73].

## 3. Embedded Systems in Healthcare

Embedded systems are electronic devices that are specifically designed to perform certain functions. They provide high levels of system integration for the development of manufacturing processes and the use of goods and services.

Embedded systems are usually composed of hardware, firmware and software. The hardware consists of the physical, electronic components that are required to fulfil the functionality of the embedded system. The main element is usually a processing unit (microprocessor, microcontroller or Digital Signal Processor) that controls the integrated circuits, such as memories, analogue-digital converters, power supplies or battery controllers. The software is a set of instructions or programs that are programmed in the processing unit to respond to specific use cases or functions of the system. The firmware is the set of instructions implemented at the processing unit to control the electronic circuitry. Firmware is considered the link between hardware and software.

Embedded systems have been transforming the healthcare industry over the last few years. An increasing number of smart devices are enabling continuous monitoring of vital signs, glucose, etc. Ref. [74] reviews different embedded solutions for monitoring vital signs. It presents several solutions based on smartwatches or even sensors integrated into textiles or lenses. These smaller and connected devices are making it easier to capture and transmit this information to healthcare centres. Article [75] presents how smart embedded systems offer secure, low-cost communication interfaces for healthcare services. Once a significant amount of data is available, this information can be post-processed using AI diagnostic algorithms to improve the results of the diagnosis. In [76], there is evidence that AI can improve the diagnosis of rare diseases. For example, pacemakers made by embedded systems are a significant breakthrough for patients with heart disease. These devices can monitor heartbeat and react to cardiac malfunctions [77]. They also register all the data so that doctors can adjust the patient’s therapy in a more efficient way.

Sensors for healthcare monitoring are usually devoted to measuring vital signs. Currently, four basic parameters are defined as vital signs [78]: blood pressure, heart rate, respiratory rate and body temperature.

According to WHO, blood pressure is defined as “the force exerted by circulating blood against the walls of the body’s arteries“ [79]. Among the solutions for blood pressure measurement, oscillometric systems that are able to analyse the vibration of the arterial wall based on the signal transduction method. In [80,81], different wearable designs based on capacitive sensors are presented. There are also auscultatory systems based on microphones that can interpret sounds during the measurement process, but due to their measurement principle, these should be used in low-noise environments [82]. There are also other methods that allow the estimation of blood pressure, an example of which is the one presented in [83], which is able to estimate blood pressure using a photoplethusmogram or the one presented in [84], which estimates blood pressure without contact using video analysis.

Heart rate, heart beeps per minute, it is commonly measured by electrocardiographs that measure the potential generated by the electrical signals that control the expansion and contraction of the heart. In [85], the author presents the design of a portable electrocardiograph. It can also be measured by optical systems that determine the heart rate by emitting a beam of light into the subchoroidal vessels and measuring the reflected light in a photo-sensor. There are many wearable developments based on the optical system as its measurement principle does not require the use of electrodes, making it suitable for these systems [86,87]. In [88], both measurement methods are compared. This study concludes that although the traditional electrocardiograph-based measurement is the most reliable, with the optical system, it is possible to measure heart rate variability with high accuracy. There are also less precise measurement systems, such as those based on videos. In [89], the author presents a system based on facial images. The author in [90] develops a system based on an infrared CMOS camera to measure heart rate.

Respiratory rate is the number of breaths per minute [91]. This parameter can be measured by an impedance spirometer that measures the variation in body resistance during breathing [92]. It is common to measure the respiratory rate through acoustic systems placed on the neck [93]. However, one of the most accurate systems is based on capnography. It measures the concentration of carbon dioxide in the patient’s airway to determine the respiratory rate [94]. Nevertheless, it is a contact-based system that cannot always be used. There are non-contact systems, such as the one presented in [95], which use a Doppler sensor placed on the ceiling of an intensive care unit. In [96], a system capable of measuring the heart rate by using imaging systems is presented.

Finally, body temperature can be measured using different methods. On the one hand, it is possible to carry out such measurements with contact thermometers. These include temperature sensors combined with predictive algorithms for fast measurement or higher sensitivity. The system presented in [97] is based on continuous temperature measurement in the ear channel and combines the reading with statistical learning algorithms for higher accuracy. In X, the author presents a smart pillow that is able to estimate body temperature using machine-learning algorithms [98]. It is also possible to measure body temperature without contact; there are several developments based on infrared measurement [99,100].

Embedded systems are not only present in monitoring medical devices; in recent years, they have been extended to all health technology categories [101]. They can be found in diagnostic (blood glucose monitors, blood INR monitors, defibrillators and digital thermometers), prognostic (PET, digital X-ray and MRI), patient management (self-test devices for remote patient monitoring) and telemedicine applications [102].

The use of embedded systems in the medical sector has become very interesting, as it provides many important advantages. These systems are considered highly customizable and controllable, as the design of both the software and the hardware is usually tailor-made for each application. Furthermore, complete design of the hardware, firmware and software allows the developer to control the system at all times. The author in [103] presents different design techniques for lightweight, re-configurable medical systems based on embedded systems. Regarding its cost, these are low-cost systems with a dedicated design that makes these systems cost-effective; this feature has opened the door to disposable or widely adopted electronic devices, such as wearable electronics. The author of [104] highlights the use of embedded systems in countries where funds are tight. Finally, due to being highly optimised systems, response times can be minimal, ensuring real-time execution. This feature is key in the medical sector as it minimises the sanitary reaction time or even the time required to dose the treatment. In [105], the importance of real-time systems in insulin pump devices is evidenced.

On the other hand, the aforementioned systems also have disadvantages or drawbacks that need to be carefully monitored when used in the health sector. These systems, especially those with accessible communication ports, can present vulnerabilities as they are susceptible to being hacked [106]. In an industrial or consumer PC-based system, it is possible to install anti-virus applications or firewalls that act as a barrier against attacks. Embedded systems are not immune to such attacks, and firewalls and anti-virus applications cannot usually be integrated into the processing unit. Therefore, it is required for the firmware/software developer and the hardware designer to implement advanced security mechanisms [107]. As it is mentioned in [108], it is considered good practice to implement secure update mechanisms, secure key storage elements, data encryption, etc.

In addition, there are many free hardware and software packages available on the web that enable almost any technician with some knowledge of electronics to implement solutions based on embedded systems. Although this may seem like an advantage, it has become a huge problem for the medical sector. These free hardware and software are not usually designed to satisfy the requirements of the healthcare sector and, therefore, do not comply with the required regulations to commercialise these solutions [109]. Usually, start-ups, due to the lack of specific technical knowledge and unfamiliarity with the medical sector requirements, develop their products based on open-source hardware and software platforms. Eventually, when they attempt to market them, they realise that a complete redesign of the developed device is required.

## 4. New Medical Product Design and Development

The development of a new medical device is a complex and resource-intensive process. The complexity of medical device development lies in compliance with the associated regulations. A weak identification of regulations and requirements or a technically optimal design without considering regulatory requirements can result in an unsuccessful new medical device development. Figure 2 presents the phases of a medical product design and development strategy. These stages are detailed in the following [110]:Feasibility: this phase identifies the market needs, clinical and regulatory aspects of the project development and economic impact.Design and Development: all the functional requirements of the development are identified in this phase, and the project plan is established. After this definition, the first prototype is developed. It is common to have design iterations during this process.Design verification: development is verified and, therefore, a functional prototype is obtained; in this stage, it is ensured that the components fulfil the established requirements and safety standards.Certification and Qualification: in this step, it is necessary to guarantee the compliance of the product with the requirements established by the accredited certification and standardisation organisms. This phase includes (i) clinical trials: depending on the category of the equipment, it is necessary to carry out clinical trials to guarantee compliance with the requirements, (ii) electrical safety tests, electromagnetic compatibility tests, etc., and (iii) achieve the product’s commercialisation acceptance.Industrialisation: during this phase, the product is transferred to production; for a successful industrialisation, this step must include quality assurance control mechanisms.Post-market surveillance: once the product is on the market, it is necessary to monitor it in order to identify problems in the field, to know the satisfaction with the use of the device and to publish software updates in response to identified vulnerabilities.

In the development of new medical products, it is mandatory to have a good quality management system [111]. Both European and U.S. regulations require a comprehensive quality management system that covers the design, development, industrialisation and post-market phases [112].

ISO 13485 defines a quality system that is recognised worldwide. Its compliance is evidence that the organisation has the required procedures established to be able to design, manufacture and maintain a medical device [113]. To demonstrate compliance with ISO 13485, medical device manufacturers or design centres can obtain the corresponding certificate. The certification of a company in ISO 13485 is not mandatory to design and develop a medical device. However, it eases the product certification process. The implementation of ISO 13485 in an organisation is not usually straightforward as it involves establishing procedures for: planning, requirements definition, design control, change management, verification and validation, documentation and transfer.

## 5. Embedded Medical Sensor Systems

An embedded medical device is usually composed of a combination of various hardware and software elements, such as processors, operating systems, memories, power supplies, user interface, communications and other peripherals (scanners, cameras, buttons and pedals). In the following sections, the different elements that make up an embedded medical device are described. In addition, the main requirements that must be met to include such components in a medical device are extracted.

Figure 3 shows the main components of an embedded medical device and the relationships between each of them.

### 5.1. Embedded Processors

Embedded processors are hardware elements designed to cope with the needs of embedded systems. They are usually low-power, low-computing-capacity but highly optimised processors. In contrast to traditional processors, they often include built-in peripherals to further optimise power consumption and cost.

Processing performance is measured in units of Dhrystome Millions of Instructions Per Second (DMIPS) [114]. Dhrystome is a computer program developed by Reinhold P. Weicker in 1984 and is used as a benchmark measurement of the overall performance of different processors. DMIPS refers to the average number of instructions executed per second, so the higher the DMIPS, the higher the processing performance of the processor.

Different data processors are available, including microprocessors (MPU) [115], microcontrollers (MCU) [116], Digital Signal Processors (DSP) [117], System on Chip (SoC) [118] and Field Programmable Gate Arrays (FPGA) [119], which are the typical ones. These are all commonly used in embedded systems; one of them or a combination of several are used to be the processing core of embedded medical devices.

The processor is key when designing an embedded system for the medical sector. When selecting a processor for a medical device, it is important to make sure that the platform has protection mechanisms, such as source code encryption, a secure storage system for encryption keys, advanced update systems, etc. [120]. Likewise, if an embedded processor is to be used in a system on a module-type platform, it is important to identify manufacturers that have ISO 13485 certification or similar. For example, Variscite, a manufacturer of SOMs, produces all of its System On Modules in compliance with ISO 13485 [121]. In addition, the supply of the processor must be guaranteed for the lifetime of the device. Medical devices, due to the regulatory requirements under which they are regulated, must pass costly tests in order to be marketed. A change in a component, such as a processor, can result in the re-certification of the device.

Furthermore, the different embedded processor manufacturers usually offer firmware/ software libraries that help to start up the different peripherals. It is also important to check that the resources offered are designed in accordance with medical device regulations or that there is sufficient information to verify and adapt these developments.

### 5.2. Embedded Operating System (OS)

The operating system is a set of programs that allows the management of hardware resources that are available on an embedded system [122]. Embedded OSs often rely on platforms with limited processing capacity, so they tend to be highly optimised. Unlike desktop operating systems, such as Windows 10 or Debian, they generally do not offer a multitude of software resources and tools. In most cases, the developer has to generate such customised libraries.

Embedded operating systems’ main characteristics are the multi-tasking capability, real-time operation and suitability for safety-critical applications, such as medical devices or automotive solutions. Some operating systems are pre-certified for critical applications facilitating the certification process of the embedded device. In addition, the operating system to be used depends on the platform on which it is going to be executed [123].

Currently, some of the operating systems that can be found pre-certified for medical devices are: Wind River [124], Integrity [125], QNX [126], Nucleus [127] or SafeRTOS [128]. All of them offer software and documentation ready to be integrated into medical devices.

In the case of operating systems that are not certified for the medical sector, it is necessary to use operating systems that allow the separation of different parts of the software. It will be key to separate safety-critical and non-critical software in order to mitigate or simplify risks during the design and development phase of the medical device [129]. Critical software is software whose malfunction could result in harm to the patient or operator.

Further, for medical systems that require real-time response, typically systems that generate alarms or are capable of dosing medication, it will be necessary to ensure that the operating system offers very low response times.

### 5.3. Memory

The memory or data storage device is used to store the embedded program or firmware as well as user data. It can be classified into volatile [130] and non-volatile memories [131,132].

In a medical device, data storage memories are especially important. These elements must contain measurement or monitorisation algorithms, device critical algorithms, as well as sensitive data.

The main characteristics of memory devices are the following:Nominal capacity: the amount of information (bytes) that a device can store.Access speed or access time: time elapsed from the moment that the memory address is provided until the data are available.Memory cycle: time elapsed between two consecutive memory accesses.Cost per bit: price per information bit.Volatility: this parameter indicates whether the stored information is lost when the power supply is cut off.Life time: some memories have a limited number of write cycles, therefore, this parameter is defined as the number of write cycles that can be performed without risk of losing information.

Similarly to embedded processors, manufacturers of medical memories must have ISO 13485 manufacturing certification, and their storage devices must be certified. Memories used in the consumer market are generally not suitable for medical devices. These elements must offer long-term availability to avoid possible re-certification of the product. In addition, elements with high reliability and security features should be chosen.

Medical device memories often contain sensitive information, so it must be guaranteed that the information is inviolable and can only be read by authorised personnel. This is usually solved through encryption mechanisms and access control with authentication.

### 5.4. Battery-Powered Devices

Medical devices can be stand-alone (battery-powered) or plug-in devices that need to be continuously connected to an external power source.

Battery-powered devices are usually powered by low-capacity lithium batteries. These batteries rely primarily on the movement of lithium ions between the positive and negative electrodes of which they are composed [133]. The negative electrode material is a fuel, so if the battery is in a state of overcharge, there is a potential risk of explosion. Further, these cells dissipate heat in a non-uniform way, which can lead to a considerable reduction of the battery’s lifetime. For this reason, lithium batteries need to have a monitoring, management and protection system.

When selecting a battery, it is necessary to consider the following parameters:Nominal capacity [134]: this parameter indicates the amount of discharged electricity from the battery under certain conditions, such as discharge rate and temperature. This capacity is defined as the maximum power that the battery can supply per hour.Battery charge and discharge rate [135]: this indicates the speed at which the battery is charged and discharged. It is defined as the charge or discharge current divided by the nominal capacity of the battery. For example, if a 4000 mAh battery is discharged at 1000 mA, its discharge rate is 0.25 C.Depth of Discharge (DOD) [136]: this parameter refers to the battery depletion rate. It indicates the percentage of the discharged capacity with respect to the nominal rating during use.State of Charge (SOC) [137]: indicates the percentage of the remaining energy compared to the battery’s nominal capacity.State of Health (SOH) [137]: refers to the state of health of the battery. This parameter evaluates the state of the battery compared to the ideal conditions. It is usually represented as a percentage value.Internal battery resistance [138]: batteries have an internal resistance as the elements that compose them are not perfect conductors. This parameter will vary with age; the higher the resistance, the higher the energy losses and, therefore, the battery will heat up more.Life cycles [139]: this refers to the number of charge and discharge cycles a battery can undergo before its capacity drops below a certain value. This parameter will depend on the quality and the materials of the battery.

The Battery Management System (BMS) [140] is responsible for controlling and managing the storage system. The BMS estimates battery charge, monitors and analyses battery health, implements safety mechanisms and manages energy control.

Usually, the BMS is composed of several electronic components. To design a battery-powered medical device, it is important to know the main blocks that will form the base of the device’s power supply system.

Medical devices may include primary or rechargeable batteries. This element is considered critical as it can overheat and/or explode, causing harm to the patient or operator [141]. In the case of batteries, there are standards such as UL 2054 or IEC 62133 that must be met when integrating a battery into a medical device. These standards will be discussed in the next section of this article.

In addition to integrating pre-certified batteries, it is necessary to take design measures to protect the system. For example, limiting the maximum charging current both by hardware and software and integrating redundant temperature sensors that monitor the battery status.

### 5.5. User Interface

The user interface is the set of peripherals and channels through which a user can communicate with a medical device. Typically, medical devices include displays, touch panels and buttons to interact with the end user.

Medical grade display or touch screens, as well as all hardware components, must be certified according to IEC 60601-1 and ISO 13485. In particular, when choosing displays for use in the medical sector, it is important to check technical characteristics such as usability with gloves, viewing angles or luminance.

Luminance (screen brightness) and contrast (light-to-dark ratio) are key. If the screen is intended to be the diagnostic platform, a wrong selection of the display can lead to an incorrect diagnosis. A display for diagnostic purposes must offer enough luminance and contrast to create at least 256 visually perceptible shades of grey [142].

Touch screens are most often used while wearing gloves, so it is necessary that the selected device supports this use case.

Finally, the viewing angle is another critical parameter; this parameter defines the maximum angle through which a good view of the screen is obtained. Often the operators view the screen from different positions as they move around the surgery or examination room. Therefore, it is important to choose screens with In-Plane-Switching (IPS) technology that maintains a good ratio of luminance and contrast at high viewing angles [143].

### 5.6. Communications

The interoperable exchange of information provides significant benefits for healthcare systems and telemedicine. On the one hand, it makes it easier for healthcare providers to have all patient data centrally available. On the other hand, patients have access to a large amount of information about their health status.

This requires interoperability between different medical devices as well as between data servers. Therefore, the standardisation of communication protocols is a basic requirement for new e-Health solutions. In this context, to overcome integration difficulties arising from the lack of standard communication interfaces and lack of homogeneity in communications, it is necessary to use standard protocols such as those defined in ISO/IEEE 11073 (X73), HL7 or POCT1-A2.

#### 5.6.1. Health Level Seven (HL7)

Health Level Seven (HL7) is a communication standard that provides resources for information exchange between different healthcare information systems. This protocol is defined and maintained by Health Level Seven International, a not-for-profit organisation working to create standards for healthcare. This organisation has a membership of more than 500 healthcare organisations in over 50 countries [144].

The aim of HL7 is to achieve interoperability between different hospital systems. This protocol is used in hospitals, medical centres, laboratories, pharmacies, emergency services, medical hardware and software manufacturers.

Currently, the HL7 standards consist of five groups of standards: HL7 Version 2 [145], HL7 Version 3 [146], CDA [147], HL7 FHIR [148] and CCOW [149].

#### 5.6.2. ISO/IEEE 11073

ISO/IEEE 11073 is a family of standards that aims to ensure interoperability between different medical devices [150]. For this purpose, there are two groups of standards: (i) Point-of-Care medical device (PoC-MDC) and (ii) Personal Health Device (PHD).

The ISO/IEEE 11073 (X73) family of standards covers the seven levels of the OSI protocol stack, providing flexibility for exchanging medical data between a Medical Device (MD) and a central system; e.g., a Compute Engine (CE). In addition, the data exchanged between the medical device and the central system can be sent to a remote-control centre for storage in the Electronic Health Record (EHR). Communication with the EHR is also regulated by ISO 13606 [151].

The X73 standard was created with the aim of its implementation in Intensive Care Units, specifically for Point-Of-Care (PoC) devices. Subsequently, it has undergone several evolutions towards new use cases, profiles, etc. Accordingly, the European Committee for Standardisation has adapted the X73 to the current technological scenario with an extension for personal healthcare (Personal Health Device, X73PHD).

#### 5.6.3. Point-of-Care Connectivity (POCT1-A2)

POCT1-A2 is an optimised communication standard for Point of Care (PoC) devices [152]. POC are medical devices that are used to obtain results in a quick way, without the need for a laboratory. Medical devices such as glucometers, coagulometers or thermometers are considered PoC devices. These are usually embedded systems with limited resources.

In 1990, the American Society for Testing and Materials (ASTM) [153] published the first specification for exchanging information between medical instruments and computer systems (ASTM E1394). This guideline was not widely deployed, so in 2002, the Clinical and Laboratory Standards Institute (CLSI) defined the Clinical Instruments and Computer Systems (LIS02-A2 [154]), which replaced the previous one. LIS02-A2 is currently known as POCT1-A2; this standard was accepted by the International Organization for Standardization (ISO) and is covered by ISO11073-90101 [155].

### 5.7. Measurement or Monitorisation Core

Embedded medical devices, specifically PoC devices, are usually oriented to monitor or characterise some parameter related to the health of patients. The functional block related to this functionality is the measurement or monitoring unit. This module is considered to be one of the most critical components of the device. Usually, this unit is made up of sensor devices (temperature, current, voltage, pressure and flow) as well as vision systems (scanners and cameras).

This component is highly dependent on the intended use of the medical device to be designed. As discussed in Section 3, there are different measurement principles for different applications. Specifically, in Section 3, the measurement methods associated with the monitoring or measurement of vital signs have been reviewed.

To design this system, it will be necessary to analyse the different measurement principles that exist to measure or monitor the required parameter. The analysis of the state-of-the-art will help to select the most accurate and appropriate measurement method depending on the device to be developed. Likewise, through this analysis, scientific evidence will be obtained to prove that the design of the device has been carried out based on previously validated scientific trials.

The monitoring and measurement subsystem is often one of the most challenging to verify. Its validation often requires testing on animals or patients, which is usually very time-consuming and costly. Therefore, this system must be designed to allow exhaustive control of the progress of the monitoring or measurement process. It is common to include different Key Performance Indicators (KPIs) that are continuously evaluating the good performance of this sub-module.

## 6. European Medical Device Regulation

The commercialisation of medical devices in Europe before 25 May 2021 was regulated by three directives: 90/385/EEC on Active Implantable Medical Devices (AIMD) [156], 93/42/EEC on Medical Devices (MD) [157] and 98/79/EC on In Vitro Diagnostic Medical Devices (IVDMD) [158].

From 26 May 2022, two regulations covering medical device commercialisation came into force. 2017/745 [3], Medical Device Regulation (MDR) and 2017/746 [159], In vitro Diagnostic medical Device Regulation (IVDR).

In practice, MDR and IVDR merely lay down minimum requirements that medical devices placed on the European market must meet. The technical details behind these directives are set out in harmonised standards developed by standardisation organisations. Following these European standards, which are defined in the Official Journal of the European Union (OJEU) [160], confers presumed conformity of the product within the legal requirements that the standard aims to cover. Despite this, the use of many of the harmonised standards is optional.

According to the MDR and the IVDR, the main harmonised standards that apply to the design and development phase of an embedded medical device are the following:IEC 60601 series—Medical electrical equipment [161];IEC 62304—Medical device software—Software lifecycle processes [162];ISO 14971—Medical devices—Application of risk management to medical devices [163];IEC 62366—Medical devices—Application of usability engineering to medical devices [164,165];ISO 13485—Medical devices—Quality management. Requirements for regulatory purposes [166];IEC 81001-5-1—Health software and health IT systems safety, effectiveness and security—Part 5-1: Security—Activities in the product life cycle [167];IEC 62133-2—Secondary cells and batteries containing alkaline or other non-acid electrolytes—Safety requirements for portable sealed secondary cells and for batteries made from them for use in portable applications—Part 2: Lithium systems [168];IEC 60086-4—Primary batteries—Part 4: Safety of lithium batteries [169].

Figure 4 shows the relationship between the typical modules of an embedded medical device and the regulations that apply to them.

Although this article reviews the requirements associated with the design and development phases of medical devices for their subsequent commercialisation in the European market in detail, the adoption of the new Medical Device Regulation (MDR) requirements brings the European regulations closer to the requirements of the FDA.

Most of the design and development phase requirements presented above apply to both the European and US markets. This is because both regulations recognise the main standards reviewed in this article: IEC 62304, IEC 62366 or ISO 14971, as harmonised standards. In addition, the FDA requires the adoption of quality management systems equivalent to ISO 13485 [170].

### 6.1. Medical Device Regulation (MDR)

The 2017/745 [3] Medical Device Regulation (MDR) defines the rules concerning the placing on the market of medical devices for use in humans. This regulation covers medical devices, accessories for medical devices and clinical investigations related to medical devices.

This regulation defines a medical device as:

“Any instrument, apparatus, appliance, software, implant, reagent, material or other article intended by the manufacturer to be used, alone or in combination, for human beings for one or more of the following specific medical purposes [...]”[3]

#### 6.1.1. Medical Device Classification According to MDR

Medical devices can be classified into different categories based on their inherent risks. The classification of the device is decided by the manufacturer, who, in the case of doubt, may consult the notified body. In Europe, risk is defined on the basis of the human body’s vulnerability to the device, the intended purpose and the duration of use. However, each regulatory authority has a different classification. According to the MDR, medical devices are classified into four categories. Each category is defined in Table 1.

#### 6.1.2. MDR Requirements for Medical Device Design and Development

The MDR covers all phases of the life cycle of a medical device [3]. This article aims to extract only the requirements associated with the design and development phases. For the fulfilment of the requirements of these phases, the MDR refers to standards that are discussed in the following sections, such as ISO 13485, IEC 60601 or IEC 62304 [171].

As a starting point, based on these regulations, it will be necessary to identify the class of the device as the scope of the different standards varies.

It will also be necessary to identify all the specific regulations that apply to the device. This article covers general standards, but there are also specific standards that must be complied with depending on the use case of the device.

### 6.2. In Vitro Medical Device Regulation (IVDR)

The 2017/746 [159] In Vitro Medical Device Regulation defines the rules for placing on the market in vitro medical devices.

According to the IVDR, the in vitro health product is defined as:

“Any medical device which is a reagent, reagent product, calibrator, control material, kit, instrument, apparatus, piece of equipment, software or system, whether used alone or in combination, intended by the manufacturer to be used in vitro for the examination of specimens, including blood and tissue donations, derived from the human body, solely or principally for the purpose of providing information [...]”[159]

#### 6.2.1. In Vitro Medical Device Classification According to IVDR

In vitro devices can be classified into four categories depending on the intended purpose of use and the risk they present. To appropriately categorise in vitro devices, Annex VIII defines seven rules. The different categories are detailed in Table 2.

#### 6.2.2. IVDR Requirements for In Vitro Medical Device Design and Development

This regulation is equivalent to the MDR but for in vitro devices. As with the MDR, this regulation refers to different standards that regulate the design and development phases.

In this case, it will also be necessary to identify the classification of the device as well as all the specific regulations that apply to it.

### 6.3. IEC 60601—Basic Safety and Essential Performance

IEC 60601 is a series of technical standards intended to ensure the safety and good performance of electrical medical devices. This series of standards published by the International Electrotechnical Commission (IEC) includes a primary or general standard (IEC 60601-1) of about 10 collateral standards (IEC 60601-1-x) that specify the general requirements which only apply to some devices; and about 80 particular standards (IEC 60601-2-x) that specify particular requirements for some medical products. The specifications defined by the particular standards prevail over those defined in the general and collateral standards. In addition, technical reports (IEC 60601-4-x) are also published, which serve as guidance for different aspects related to medical devices.

The general standard, the IEC 60601-1 [161], defines the general requirements for basic safety and essential performance. This standard appeared in 1977 with its first edition (Ed. 1). In 1988, a second version (Ed. 2) was generated, which focused on ensuring safety in the vicinity of the patient. The third edition (Ed. 3) is currently in force and has been since 2005. In this edition, protection has been extended to patients and equipment operators. In addition, a major review of the standard was carried out in 2012, which clarified several ambiguities generated by the evolution of technology. A new edition of the standard is expected to be published in 2024 [172].

Different editions of the standards are widely accepted in many countries, all of Europe (Ed. 3.1), Canada (Ed. 3.1), USA (Ed 3.1), Japan (Ed. 3.1), China (Ed. 2), Brazil (Ed. 3.1), South Korea (Ed. 3.1) and Taiwan (Ed. 2) recognise the IEC 60601 standards.

#### Safety Requirements for Medical Device Design and Development

First of all, it will be necessary to identify the classification of the device that depends on the type and degree of protection against electric shocks.

It is also required to identify which collateral and particular standards apply to the device in order to extract specific technical requirements such as the performance that the device must have regarding electrostatic discharge (ESD), radiated and conducted immunity or during dropping tests.

Once the requirements have been identified, the standard defines that it will be necessary to perform tests to ensure the functional safety of the device.

### 6.4. IEC 62304—Medical Device Software

The IEC 62304 [162] standard is published by the International Electrotechnical Commission (IEC). It provides processes to develop software for medical devices. For that, it defines lifecycle processes with activities and tasks necessary for the safe design, development and maintenance of medical device software. This standard references ISO 14971 on medical device risk management, and both are aligned.

The first version of this standard was published in 2006; later an amendment was generated in 2012; this version is known as edition 1.1. Currently, edition 2 of the standard is being drafted and is planned to be released in 2023; it is expected that this version will also include software equipment that is not considered to be a medical device in its processes [173].

IEC 62304 is applied to all medical devices in which software is available as a core or relevant part of the medical device. It also applies when the device is composed entirely of software components.

This standard covers the software development and maintenance phases but does not cover the final validation and dissemination of the device.

#### 6.4.1. Software Safety Classification

The manufacturer must classify each software system according to its safety class (A, B or C). This classification depends on the possible effects on the patient or other people in the vicinity.

Class A: software that cannot cause injury or harm to health;Class B: software can cause injury, but not serious;Class C: it is possible to cause death or serious injury.

In case the use of hardware or other mechanisms can mitigate or reduce the risk of injury, the class of the software can be reduced. In addition, the risk analysis performed according to ISO 14971 has to include the classification of each software system.

#### 6.4.2. Software Requirements for Medical Device Design and Development

First of all, the software classification of the device must be identified, as depending on this, the scope of other requirements may vary.

It is also necessary to establish a software development plan that covers all phases of the design and development process.

The software design process must start with the extraction of software requirements. These must be first transformed into an architectural design and then into a detailed design.

Once the design is in place, the software must be implemented, and the different components and their integration must be verified. The results of the verification must be recorded.

It will also be necessary to perform a system check to ensure that the requirements are met. Once this verification has been carried out and the required documentation has been generated, the software can be released.

### 6.5. ISO 14971—Medical Device Risk Management

ISO 14971 [163] provides a framework for managing the risks associated with a medical device. For this purpose, it defines patient, operator and human risk management processes. More specifically, this standard was developed to enable manufacturers of medical devices to develop and maintain a risk management system.

The ISO 14971 standard is based on EN 1441, Medical devices—Risk Analysis, published by the European Committee for Standardization (CEN) in 1997 and the ISO 14971-1, Medical devices—Risk management—Part 1: Application of risk analysis, published in 1998. The first edition of ISO 14971 was published in 2000, the second edition in 2005 and the third edition in 2019. Since 2021, this standard has been harmonised with two European Regulations, the 2017/745 (MDR) and the 2017/746 (IVDR).

This standard defines processes for a medical device manufacturer to identify, estimate, evaluate and control the hazards associated with their product. This standard applies to medical devices, including in vitro diagnostic kits. In addition, ISO 14971 applies to all stages of the life cycle of a medical device. This standard does not apply to clinical decision-making and does not define acceptable levels of risks.

Although the standard does not require the manufacturer to have a quality management system such as ISO 13485, risk management can be part of a quality management system.

#### Risk Management Requirements for Medical Device Design and Development

A risk analysis of all components of the system will be necessary. Following this analysis, the risks shall be assessed.

In case there is a risk that is not acceptable to the manufacturer, control measures need to be put in place.

The risk analysis is considered an iterative process as it shall be performed again and again until the residual risk, that is, the risk that is present in the device after applying the control measures is considered acceptable by the manufacturer.

### 6.6. IEC 62366—Medical Device Usability

The IEC 62366 standard refers to usability engineering for medical devices. This standard is a process-based standard to allow medical device manufacturers to design products with high usability.

In recent years, the number of medical devices for patient observation and treatment with user interfaces has been increasing. The integration of these new functionalities has increased the number of errors in their use. The lack of design simplicity or the difficulty in learning how to use them are often the source of these errors. Usability engineering, the design of a user interface according to processes that guarantee adequate safety of the medical product, is the key. Therefore, this standard aims to minimise the risks related to user interface design in medical equipment.

This standard has two parts, IEC 62366-1 [164] and IEC 62366-2 [165]. Its first version was published in 2007 by the IEC, the International Electrotechnical Commission. In 2015 it was updated, resulting in IEC 62366-1; Application of usability engineering to medical devices. In 2016, IEC 62366-2, Guidance on the application of usability to engineering to medical devices, was generated. The requirements and tasks to be fulfilled to achieve a usable medical device are defined in IEC 62366-1. IEC 62366-2 is only a guide for the correct application of IEC 62366-1.

The IEC 62366 standard is accepted in both the European Union and the United States, so manufacturers from these markets can rely on this specification to develop their devices.

IEC 62366 specifies the process for analysing, specifying, designing, verifying and validating the usability of a medical device under normal conditions of use, according to specification and documentation of use. This standard is intended to mitigate the risks caused by usability problems of a device. IEC 62366 can be used to identify, but not to assess or mitigate, risks associated with the abnormal use of the device, that is, when it is not used in accordance with the device’s instructions for use. In this case, the assessment is performed according to ISO 14971. This standard does not apply to clinical decision-making regarding medical devices.

This standard is oriented to minimise the risks related to interface design; therefore, it makes reference to ISO 14971, specifying that in the case of following the processes defined in this standard unless there is objective evidence, the residual risks that appear after the execution of the defined processes are acceptable.

#### Usability Requirements for Medical Device Design and Development

The usability requirements are based on a risk analysis of use. For this, it will be necessary to define the use case of the device and perform a risk analysis of its use.

To assist in the design and development phases, this standard requires summative and formative evaluation.

The formative evaluation aims to explore the strengths, weaknesses and the unanticipated use errors of the user interface. In contrast, summative evaluation is intended to determine whether the interface is safe to use.

The formative evaluation plan must include the acceptance criteria, usability objectives, test environment, methods and definition of the used techniques. The summative evaluation plan should include acceptance criteria, test environment, methods and techniques, participants and satisfaction rates.

### 6.7. ISO 13485—Medical Device Quality Management Systems

ISO 13485 [166] is a standard that specifies the requirements of a quality management system for medical device manufacturers.

This standard was published in 1996 by the International Organization for Standardization (ISO); this first version was based on ISO 9001:1994. In 2003, after the update of ISO 9001, ISO 13485 was updated. The next update was made in 2012. At present, the version published in 2016 is in force and is based on ISO 9001:2008.

This process-based standard covers all stages of the product life cycle: design, development, production, storage, distribution, installation, technical support, deinstallation and final disposal of the product. It also covers the design, development and service supply related to medical devices. ISO 13485 can be adopted by both medical device manufacturers and also by their suppliers.

This standard is the main quality system for medical devices in Europe, Canada and Australia. It also serves as the basis for compliance with quality systems in other countries, such as Japan, Korea and Brazil. It has recently been published that the FDA intends to use ISO 13485 as the basis for its quality systems legislation [174].

ISO 13485 is based on ISO 9001. However, this standard includes some particular requirements for organisations involved in the life cycle of medical devices and excludes some requirements of ISO 9001 that are not appropriate as regulatory requirements.

This standard does not include specific requirements for other management systems, such as environmental management, health and safety at work, financial management, etc. However, it is possible to align this quality management system with other existing ones.

This standard specifies the requirements that a quality management system must meet when an organisation needs to demonstrate its capability to provide medical devices and related services in compliance with applicable customer and regulatory requirements.

ISO 13485 applies to organisations that are involved in one or more stages of the life cycle of a medical device. It may also be used by suppliers providing parts of the product to such organisations.

In the case where some applicable processes are carried out outside the medical device manufacturer, it is the responsibility of the manufacturer to incorporate these processes into its quality management system through process monitoring, maintenance and control.

#### Quality Management Requirements for Medical Device Design and Development

This standard requires a plan for product development.

Specifically, the design and development phases it specifies the need to plan the process, define the inputs and outputs of the design and carry out systematic design reviews, verification and validation. It also contemplates the transfer phase to manufacturing.

It also details the importance of controlling component purchases and measuring equipment during the prototyping and production phases.

According to ISO 13485, it will also be necessary to document procedures, reviews and controls.

### 6.8. IEC 81001-5-1—Medical Device Cybersecurity

Medical devices are more and more connected to allow an agile exchange of data between the device and the hospital network. Currently, many medical devices are not designed according to cybersecurity requirements, as regulations were not directly addressing this potential risk.

In Europe, the Medical Device Directive (93/42/EEC) [157] published in 1993 merely featured a sentence that indirectly referred to cybersecurity aspects. It was not until the new Medical Device Regulation (2017/745), in force since 26 May 2021, that direct reference was made to cybersecurity.

Specifically, the list of harmonised standards for the MDR refers to three different cybersecurity standards, for which the date of adoption is May 2024.

IEC 80001-1 [175]: safety, effectiveness and security in the implementation and use of connected medical devices or connected health software—Part 1: application of risk management, which is oriented towards security criteria for networks in which medical devices are embedded.IEC 81001-5-1 [167]: health Software and health IT systems safety, effectiveness and security—Part 5-1: Security: activities in the product lifecycle, focused on the cybersecure design of medical devices. It defines the cybersecure development life cycle.IEC/TR 60601-4-5 [176]: medical electrical equipment—Part 4-5: guidance and inter-pretation—Safety-related technical security specifications. It is a technical report that complements IEC 81001-5-1 with additional security requirements.

Since 2014, the FDA has published several guidelines to ensure cybersecurity in its equipment. In 2018, the FDA recognised UL 2900 as the first cybersecurity standard [177]; however, in future revisions of the guidance, this standard has lost momentum.

The latest guidance published by the FDA in April 2022, “Cybersecurity in Medical Devices: Quality System Considerations and Content of Premarket Submissions”, reduces UL 2900 to a cybersecurity testing guideline. On the other hand, it makes reference to ISA 62443-4-1 as a possible framework for developing the software lifecycle in a cyber-secure way [178].

IEC 81001-5-1 is based on ISA 62443-4-1, thereby providing a common reference for both markets. However, there are still some differences in the scope of some concepts presented in the FDA guidance and IEC 81001-5-1. Therefore, it is expected that in the coming years, IEC 81001-5-1 will become the reference standard for medical device cybersecurity for Europe and international markets.

This standard defines the requirements of the software development and maintenance lifecycle to be compliant with IEC 62443-4-1 (Security for industrial automation and control systems—Secure product development lifecycle requirements).

Based on IEC 62443-4-1, this standard aims to increase the cybersecurity of healthcare software. To this end, it sets out specific activities and tasks.

IEC 81001-5-1 applies to health software; that is, software which is part of a medical device, software as part of hardware specifically intended for health use, software as a medical device and software only produced for health use.

#### Cybersecurity Requirements for Medical Device Design and Development

This standard completes the requirements already defined in IEC 62304 from a cybersecurity point of view.

It covers the same design and development phases: planning, software requirement analysis, architectural design, detailed design, software unit implementation and verification, software integration testing, software system testing and software release. In this case, this standard emphasises the importance of specifying requirements, design and tests to ensure that the medical device to be developed is cybersecure. To this end, it recommends the use of software coding standards or good cyber-safe design practices, such as defence-in-depth and effective software segregation [179].

### 6.9. Batteries Regulation

More and more medical devices, especially portable embedded devices, include batteries. For regulating the use of batteries in medical devices, different standards exist [180]. These must be considered when selecting the battery to be used. The following standards are identified for the use of batteries in the European medical device market:IEC 62133: secondary cells and batteries containing alkaline or other non-acid electrolytes—Safety requirements for portable sealed secondary cells and for batteries made from them for use in portable applications. This standard applies to devices with rechargeable batteries.IEC 60086-4: primary batteries—Safety of lithium batteries. This standard applies to devices with non-rechargeable batteries.

The following certifications are required for FDA-regulated devices:UL 2054: UL Standard for Safety Household and Commercial Batteries. Applies to devices with rechargeable and non-rechargeable batteries.UL 1642: UL Standard for Safety Lithium Batteries. Applies to devices with rechargeable and non-rechargeable batteries.

In addition, it is important to mention that the shipment of lithium batteries is considered dangerous and must be properly tested and packaged. The transport of lithium batteries is regulated by the UN (United Nations) in the UN Manual of Tests and Criteria, Sub-section 38.3. This standard defines environmental, mechanical and electrical requirements for all lithium cells and batteries.

## 7. Conclusions and Future Work

The research work presented in this article provides the technological and regulatory aspects to be considered for the introduction of Embedded Sensor Systems for Healthcare. This article shows that the healthcare sector requires more and more devices to help healthcare professionals diagnose and treat patients. The development of these devices is not an easy task as these devices are regulated under strict regulations.

Therefore, there is an increasing need to develop new medical devices with sophisticated technical solutions that allow the healthcare sector to evolve. The emerging new technologies have enabled the emergence of start-ups that aim to introduce new devices into the market, with no need to invest large resources in their development. The development of new medical devices based on embedded systems offers important opportunities for the health sector. However, it is necessary to consider the peculiarities of these systems in order to develop a medical device successfully.

New product development in the medical sector requires the knowledge to identify applicable requirements and regulations and to establish a quality management system (ISO 13485) to ensure quality during the whole life cycle of the product. Depending on the company’s nature, it can be difficult to successfully develop new medical devices. Although the concept of innovation and transformation is often well understood, the lack of awareness of the relevance of analysis, design and verification phases often causes the death of many medical device concepts along the market path.

Once the technical and regulatory aspects of medical devices with Embedded Sensors have been broadly defined, as future research, a methodology for the design and development of embedded medical devices will be proposed. On the one hand, this methodology must contemplate technical, regulatory and methodological requirements. On the other hand, it must minimise the development risk and maximise investment efficiency. Finally, it must be suitable for companies with no experience in the sector.

## Figures and Tables

**Figure 1 sensors-22-09917-f001:**
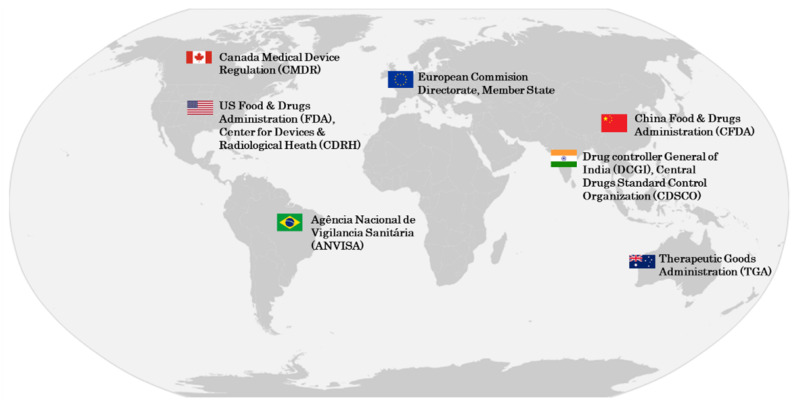
Regulatory authorities around the world.

**Figure 2 sensors-22-09917-f002:**
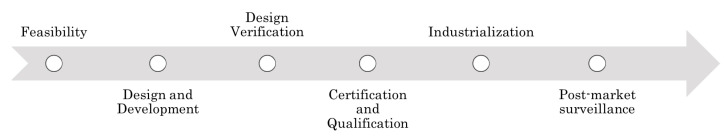
The phases of a medical product design and development strategy.

**Figure 3 sensors-22-09917-f003:**
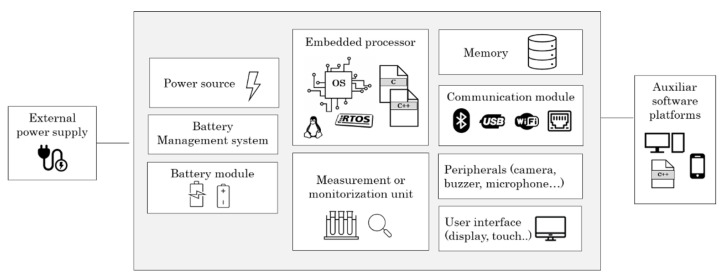
Typical embedded medical device block diagram.

**Figure 4 sensors-22-09917-f004:**
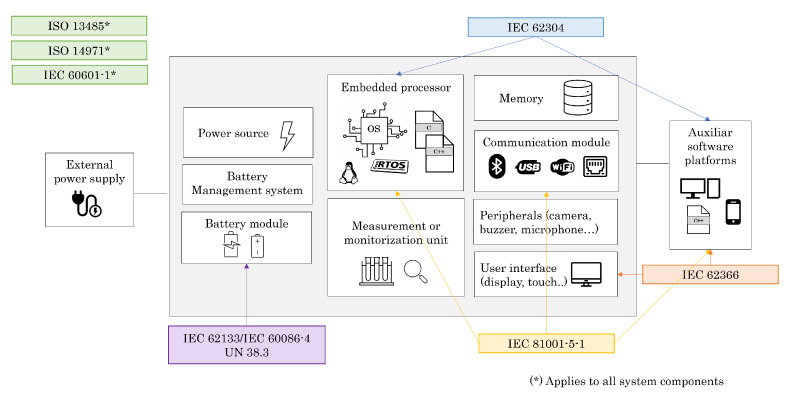
Embedded medical device modules and its regulations.

**Table 1 sensors-22-09917-t001:** Medical Device classification according to MDR.

Class	Risk Level	Device Type	Examples
I	Low	Non-invasive, sterile and for measurement	Insoles for feet and adhesive plasters
IIa	Medium	Short-term invasive device with no significant effect on organisms and fluids	Scalpel, ultrasound machine and feeding probe
IIb	Medium—High	Long-term invasive device and significant effect on organisms and fluids	Intraocular lenses, external defibrillator, dialyser and X-ray
III	High	Can be potentially life-threatening, fully absorbed in the human body, implantable and medicine of animal source	Can be potentially life-threatening, fully absorbed in the human body, implantable and medicine of animal source

**Table 2 sensors-22-09917-t002:** In vitro medical device classification according to IVDR.

Class	Individual Risk Level	Public Health Risk	Examples
A	Low	Low	Specimen receptacles, laboratory instruments and buffer solutions
B	Medium	Low	Blood chemistry and pregnancy tests
C	High	Medium	Oncological markers and dangerous infectious diseases
D	High	High	Blood safety and high-risk infectious diseases

## Data Availability

Not applicable.

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
