# Peer review of "Embedded Sensor Systems in Medical Devices: Requisites and Challenges Ahead"

_sensors, 2022, doi:10.3390/s22249917_

Round 1

Reviewer 1 Report

This paper summarizes the challenges to be faced when designing Embedded Sensor Systems for the medical sector. The overall organization is clear and concise. 

Suggestions: 

1. Line 96:. "The WHO (World Health Organisation) estimates that between 2015 and 2050,..... ". Where is the information source, news, or research?

2. Line 110 ,115:Why the authors make sure that "Optimization of data management, Artificial vision systems" is the advances and developments that are revolutionizing the medical sector? Does any research support this?

3. Line 167: Some research to support the "Embedded systems in healthcare" is necessary. In the section, the paper does not cite any research.

4. Line 416: Any research supports this: "This component is highly dependent on the intended use of the medical device to be designed"? In the section, the paper does not cite any research.

5. The paper could summarize some published embedded Sensor Systems in medical devices.

6.  This is research not to write a book. It needs more references to support the views of the paper proposed. Not only the statements for the authors.

Author Response

Response to Reviewer 1 Comments

Point 1: Line 96:. "The WHO (World Health Organisation) estimates that between 2015 and 2050,..... ". Where is the information source, news, or research?

Response 1: References are added

Point 2: Line 110 ,115:Why the authors make sure that "Optimization of data management, Artificial vision systems" is the advances and developments that are revolutionizing the medical sector? Does any research support this?

Response 2: References and examples are added to support the statement.

Point 3: Line 167: Some research to support the "Embedded systems in healthcare" is necessary. In the section, the paper does not cite any research.

Response 3: References and examples are added to support the statement.

Point 4: Line 416: Any research supports this: "This component is highly dependent on the intended use of the medical device to be designed"? In the section, the paper does not cite any research.

Response 4: The meaning of this statement has been clarified and supporting references are detailed.

Point 5: The paper could summarize some published embedded Sensor Systems in medical devices.

Response 5: An extensive literature review is carried out and different developments of embedded systems are presented. In addition, it goes in depth into sensors for monitoring vital signs.

Point 6: This is research not to write a book. It needs more references to support the views of the paper proposed. Not only the statements for the authors.

Response 6: The literature review is extended.  In addition, the extraction of requirements from standards as well as technological components that form embedded systems is added as a differential with respect to existing articles.

Reviewer 2 Report

The article titled: Embedded Sensor Systems in Medical Devices: The Challenges Ahead presents a  summary of the challenges to be faced when designing Embedded Sensor Systems for the medical sector.

The introduction presents relevant statistics numbers where they do not have citations. Therefore, it is a risk of plagiarism of this information. 

Lines 38-48 seem to be copied from reference 2, which is not acceptable due to the size of the paragraph. Please, check the citation rules. 

Section 2 lacks citations.

Sections 3, 4, and 5 present general information and do not mention the requirements to build an electronic medical system. 

Section 6 is interesting. However, it is a simple description of each regulation law without deeply analyzing how it affects the electronic system design. 

In general, the article does not show novelty, and it is a description of several regulation laws. Also, it is necessary to check the citations carefully to avoid plagiarism.

Author Response

Response to Reviewer 2 Comments

Point 1: The introduction presents relevant statistics numbers where they do not have citations. Therefore, it is a risk of plagiarism of this information.

Response 1: References are added to support the data presented.

Point 2: Lines 38-48 seem to be copied from reference 2, which is not acceptable due to the size of the paragraph. Please, check the citation rules. 

Response 2: It is reformatted according to the citation rules.

Point 3: Section 2 lacks citations.

Response 3: References and examples are added.

Point 4: Sections 3, 4, and 5 present general information and do not mention the requirements to build an electronic medical system. 

Response 4: Specific requirements to be fulfilled by each subsystem of the embedded medical device are defined.

Point 5: Section 6 is interesting. However, it is a simple description of each regulation law without deeply analyzing how it affects the electronic system design. 

Response 5: instead of only providing a general definition of the applicable standards, a section is added at the end of each standard where specific regulatory requirements are extracted.

Point 6: In general, the article does not show novelty, and it is a description of several regulation laws. Also, it is necessary to check the citations carefully to avoid plagiarism.

Response 6: An extensive literature review is carried out and different developments of embedded systems are presented. In addition, it goes in depth into sensors for monitoring vital signs. Finally, the extraction of requirements from standards as well as technological components that form embedded systems is added as a differential with respect to existing articles.

Reviewer 3 Report

The article is about embedded sensors systems in the medical field. In the study, first of all, these systems were introduced roughly, and then the subject of "European Medical Device Regulation" was mainly discussed. From this point of view, I think the choice of title will be disappointing for the reader. When the reader looks at the article title, he or she will look for an answer to the question "how are sensors embedded" or "what are embedded sensor systems". In the content of the article, sensors, embedded sensors or embedded systems are presented very briefly.

In my opinion, this issue is critical information in terms of commercialization and has a serious importance in terms of literature. In this part, it is good that the reader does not get bored with the article. But it would be nice to offer branches, differences, concepts or comparisons as a guide for researchers and those who want to commercialize. It would also be good to provide a comparison/similarity with other regulations globally.

The work is quite valuable in my opinion, but it is obvious that it is not fully mature. For this reason, it is essential that the authors examine the following similar review articles and present their differences from these articles.

https://doi.org/10.1007/s44174-022-00002-7

https://doi.org/10.3390/gels7040207

https://doi.org/10.1109/RBME.2020.3033930

https://doi.org/10.3389/fbioe.2019.00313

Author Response

Response to Reviewer 3 Comments

Point 1: From this point of view, I think the choice of title will be disappointing for the reader. When the reader looks at the article title, he or she will look for an answer to the question "how are sensors embedded" or "what are embedded sensor systems". In the content of the article, sensors, embedded sensors or embedded systems are presented very briefly.

Response 1: The title is adapted, the existing embedded sensors as well as the requirements they have to fulfil to be a medical device are discussed in more detail.

Point 2: In my opinion, this issue is critical information in terms of commercialization and has a serious importance in terms of literature. In this part, it is good that the reader does not get bored with the article. But it would be nice to offer branches, differences, concepts or comparisons as a guide for researchers and those who want to commercialize. It would also be good to provide a comparison/similarity with other regulations globally.

Response 2: A brief comparison is added indicating which regulations are applicable in both the European and US markets. In addition, instead of only providing a general definition of the applicable standards, a section is added at the end of each standard where specific regulatory requirements are extracted.

Point 3: It is essential that the authors examine the following similar review articles and present their differences from these articles.

Response 3: An extensive literature review is carried out and different developments of embedded systems are presented. In addition, it goes in depth into sensors for monitoring vital signs. Finally, the extraction of requirements from standards as well as technological components that form embedded systems is added as a differential with respect to existing articles.

Round 2

Reviewer 1 Report

Suggestions: 

1. Line641:646 :.  seem to be copied from "Sastri, V. R. (2021). Plastics in medical devices: properties, requirements, and applications. William Andrew.". and the research did not cite the journal, which is not acceptable due to the size of the paragraph. Please, check the citation rules.

2. section 5.4 Battery powered devices lacks citations.

3. section 6. European Medical Device Regulation lacks citations.

4. section 6.1.2. MDR requirements for medical device design and development lack citations.

5. "section 6.3.1. Safety requirements for medical device design and development,6.4.2. Software requirements for medical device design and development, 6.5.1. Risk management requirements for medical device design and development,6.5.1. Risk management requirements for medical device design and development

6.5.1. Risk management requirements for medical device design and development" The research just added and present general information and do not mention the requirements to build an electronic medical system. they lack citations and tell us why? Not each part of software engineering is necessary.

Author Response

Response to Reviewer 1 Comments

Point 1: Line641:646 :.  seem to be copied from "Sastri, V. R. (2021). Plastics in medical devices: properties, requirements, and applications. William Andrew.". and the research did not cite the journal, which is not acceptable due to the size of the paragraph. Please, check the citation rules.

Response 1: It is reformatted according to the citation rules.

Point 2: section 5.4 Battery powered devices lacks citations

Response 2: References are added.

Point 3: section 6. European Medical Device Regulation lacks citations.

Response 3: References are added.

Point 4: section 6.1.2. MDR requirements for medical device design and development lack citations.

Response 4: References are added.

Point 5: "section 6.3.1. Safety requirements for medical device design and development,6.4.2. Software requirements for medical device design and development, 6.5.1. Risk management requirements for medical device design and development" The research just added and present general information and do not mention the requirements to build an electronic medical system. they lack citations and tell us why? Not each part of software engineering is necessary.

Response 5: Each section seeks to draw out the regulatory requirements with which an embedded medical device designer must comply. In these sections, requirements are extracted as documentation or aspects to be taken into account in a general way, in order to serve as a guide for designers of embedded medical devices. Currently there are publications where specific aspects are detailed for a particular application e.g. thermometers, X-ray machines... In such cases more specific technical requirements are defined, e.g. EMC levels to be fulfilled etc. This is not feasible in this article as these parameters depend on the type of device, its intended use, the specific technology used... Therefore, the aim of this article is to fill the gap that exists in defining requirements or steps to follow when a designer is trying to design an embedded sensor system for the medical sector.

Therefore, in the subsections of chapter 5 there are aspects to be taken into account for the most common technology components of these systems and in chapter 6 the regulatory aspects.

In order to further clarify the contribution of this article, a new comparison of the contribution of this article with other existing articles is added.

Reviewer 3 Report

The corrections made by the authors are sufficient. However, I strongly recommend emphasizing the differences/scientific contribution from existing review articles. It is essential that the authors examine the following similar review articles and present their differences from these articles.

https://doi.org/10.1007/s44174-022-00002-7

https://doi.org/10.3390/gels7040207

https://doi.org/10.1109/RBME.2020.3033930

https://doi.org/10.3389/fbioe.2019.00313

Author Response

Response to Reviewer 3 Comments

Point 1: The corrections made by the authors are sufficient. However, I strongly recommend emphasizing the differences/scientific contribution from existing review articles. It is essential that the authors examine the following similar review articles and present their differences from these articles.

https://doi.org/10.1007/s44174-022-00002-7

https://doi.org/10.3390/gels7040207

https://doi.org/10.1109/RBME.2020.3033930

https://doi.org/10.3389/fbioe.2019.00313

Response 1: Included, lines 90 – 106
